# BEYOND DIRECTED ACYCLIC COMPUTATION GRAPH WITH CYCLIC NEURAL NETWORK

## ABSTRACT

This paper investigates a fundamental yet overlooked design principle of artificial neural networks (ANN): We do not need to build ANNs layer-by-layer sequentially to guarantee the Directed Acyclic Graph (DAG) property. Inspired by biological intelligence, where neurons form a complex, graph-structured network, we introduce the transformative Cyclic Neural Networks (Cyclic NN). It emulates biological neural systems' flexible and dynamic graph nature, allowing neuron connections in any graph-like structure, including cycles. This offers greater flexibility compared to the DAG structure of current ANNs. We further develop the Graph Over Multi-layer Perceptron, the first detailed model based on this new design paradigm. We experimentally validate the advantages of Cyclic NN on widely tested datasets in most generalized cases, demonstrating its superiority over current layer-by-layer DAG neural networks. With the support of Cyclic NN, the Forward-Forward training algorithm also firstly outperforms the current Back-Propagation algorithm. This research illustrates a transformative ANN design paradigm, a significant departure from current ANN designs, potentially leading to more biologically similar ANNs.

## 1 INTRODUCTION

Artificial intelligence (AI) has reshaped our daily lives and is expected to have a much greater impact in the foreseeable future. Lying behind the most profound AI applications (Silver et al., 2017; OpenAI, 2023; Ramesh et al., 2021; Jumper et al., 2021), artificial neural networks (ANN) such as multi-layer perception (MLP) (Rumelhart et al., 1986), convolution neural network (CNN) (LeCun et al., 1995) and Transformer (Vaswani et al., 2017) are designed specifically for different domains to fit the training data. Regardless of the structure, they are stacked layer-by-layer to form deep ANNs for greater learning capacity. It has been a *de facto* practice until now that data is first fed into the input layer and then propagated through all the stacked layers to obtain the final representations at the output layer. This paper seeks to answer a fundamental question in ANNs: "Do we really need to stack neural networks layer-by-layer sequentially?"

To answer this question, let's first examine the evidence from biological intelligence (BI). Neuroscientists have studied the biological neurons for decades. The connectome of C. elegans is the most thoroughly studied biological neural system, and biologists depicted the most detailed connection between 302 biological neurons (White et al., 1986; Cook et al., 2019) as shown in Figure 1(a). Rather than being stacked layer-by-layer, all the neurons form a complicated connection graph, where each can connect to several other neurons within the system. We cannot even determine which neuron serves as the input/output within the neural system to process information. The same findings have also been observed in the latter more complicated neural systems, such as the biology neural connectome of drosophila larva (Winding et al., 2023), zebrafish (Brooks et al., 2022), mouse (Sporns & Bullmore, 2014) and the human brain (Shapson-Coe et al., 2024). Observed biological intelligence exhibits graph-structured, flexible, and dynamic neural systems, which are apparently different from the current layer-by-layer ANNs we build nowadays, as depicted in Figure 1(b).

The learning rules actually cause the difference in the neural system structure between BI and AI. The Hebb's Rule (Hebb, 2005), depicted as "Neurons that fire together wire together", is recognized as the fundamental learning way of biological neurons. The Spike-Timing-Dependent Plasticity (STDP) learning is then proposed to further consider the relative spiking time of pre-synapse and post-synapse neurons. Both learning rules of BI are localized, *i.e.*, the learning occurs on each neuron within its local influence scope. The localized learning rules grant the flexibility of each neuron on its connections to other neurons, which leads to

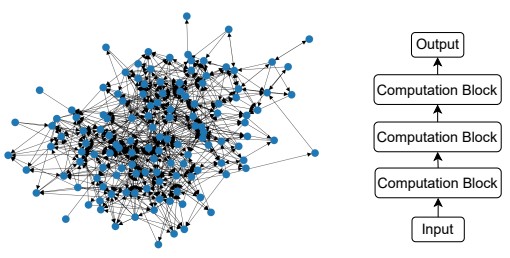

(a) Connectome of BI (C. elegans) (b) Computation structure of AI

Figure 1: Neuron connection between Biology Neural Network and Artificial Neural Network

the complicated graph-structured BI system. Conversely, for AI systems, the backward propagation (BP) algorithm (Rumelhart et al., 1986) has dominated the training of ANNs. Data is fed into the ANNs from the input layer, forward propagates layer by layer to the last layer, calculates a global loss for the whole ANN based on the ground-truth labels, and then reversely backward propagates the error signals layer by layer to the input layer. In this procedure, ANNs are trained by a global loss function, and the ANNs must guarantee the error from global loss can be back-propagated layer by layer. This requirement prevents current ANNs from forming cycles to ease gradient back-propagation. Current ANNs are nearly all DAG structured. To mitigate the biological implausible nature of the BP algorithm, the forward-forward (FF) algorithm (Hinton, 2022) is recently proposed to train ANNs. FF algorithm constructs good/bad samples and computes a loss function on each layer to differentiate between these samples. Similar to Hebb's Rule and STDP learning, the FF algorithm is a localized learning method. These advancements have allowed the training of ANNs to no longer rely solely on layer-by-layer back-propagation to design non-DAG-structured Cyclic Neural Networks.

Cyclic NN distinguishes itself from the current layer-by-layer ANNs in several aspects. 1) More flexible neuron connections. Cyclic NN greatly increases the design space of ANNs beyond the DAG structure. In Cyclic NN, the information flow is not as unidirectional as in DAG. Former neurons can also adjust based on the information encoded by the latter neurons, which largely enhances information communication within the network. The flexible connection design also makes Cyclic NN more like the biological neural system. 2) Localized training. Instead of current dominating global loss-guided BP-based training, Cyclic NN is based on localized training, *i.e.*, each neuron is optimized with its own local loss function. There is no gradient propagating between neurons. Localized training has its unique advantages. It frees the need to build DAG dependency between neurons, which is the bedrock of supporting cycles within the network. Also, each neuron is optimized independently without waiting for gradients from the latter layers. 3) Computational neuron. Different from current ANNs that a neuron is considered as a $d$ dimension to 1 dimension vector mapping; the neuron within Cyclic NN is considered the computational neuron with greater computation capacity because it is the optimization unit to fit the local task, which requires more parameters. This paper uses a linear layer to parameterize each computational neuron to fit the local classification task. It is also evident by the study of biological neuron (Beniaguev et al., 2021), which empirically proves the learning capacity of a biological neuron is much larger than a $d$ dimension to 1 dimension vector mapping function as the neuron defined within current ANNs. We take this observation and propose the computational neuron in Cyclic NN with more capable computation to fit the local optimization task. In summary, our contributions can be summarized as follows:

• Conceptually, we compare BI and AI to investigate a fundamental yet overlooked design principle: We do not need to satisfy the DAG constraint when designing ANNs.

• Methodologically, we propose the transformative Cyclic NN, a novel ANN design paradigm that supports a much more flexible connection between neurons, which discards current directed acyclic computation graph constraints.

- We test the novel design paradigm on the most generalized case and propose Graph Over Multi-layer Perceptron, the first detailed model based on Cyclic NN.

- Experimentally, we demonstrate the advantage of the proposed Cyclic NN on widely tested datasets. At the same time, we are the first to beat the current dominating BP training using the FF training algorithm by the supported flexible network design proposed in this paper.

## 2 CYCLIC NEURAL NETWORK

One Cyclic NN model is one graph $\mathcal{G} = (\mathcal{V}, \mathcal{E})$, where $\mathcal{V} = (N_1, N_2, ..., N_{|\mathcal{V}|})$ is the computational neuron set and $\mathcal{E} = (S_1, S_2, ..., S_{|\mathcal{E}|})$ is the synapse set denoting the connections among neurons. Similar to the BI system, $N_i$ ($\forall i \in \{1, 2, \cdots, |\mathcal{V}|\}$) is the neuron that tackles the detailed computation, and $S_j$ ($\forall j \in \{1, 2, \cdots, |\mathcal{E}|\}$) is the synapse that propagates information between computational neurons. In the Cyclic NN, computational neurons can be connected flexibly in any way, like the BI system.

### 2.1 COMPUTATIONAL NEURON

Computational neuron $N$ acts as the computation/optimization unit in Cyclic NN. Different from the neuron in current ANNs, which indicates a $d_{\text{input}}$ to 1 mapping, we grant $N$ with stronger computation power as it is the optimization unit to fit the local task. It is motivated by the research that proves the computation power of a single biological neuron is similar to an MLP (Beniaguev et al., 2021). In Cyclic NN, $N$ is parameterized by a function $f_N^{\mathbb{R}^{d_{\text{in}}^N} \to \mathbb{R}^{d_{\text{out}}^N}}(\mathbf{h}_{\text{in}}^N) = \mathbf{h}_{\text{out}}^N$ that maps from a $d_{\text{in}}^N$-dimensional representation $\mathbf{h}_{\text{in}}^N$ to a $d_{\text{out}}$-dimensional representation $\mathbf{h}_{\text{out}}^N$. $d_{\text{in}}^N$ is the input dimension that is decided by the output of its pre-synapse neurons, and $d_{\text{out}}^N$ is the output dimension of $N$. Similar to the biological neurons, each computational neuron functions as a computation/optimization unit. The computational neuron is also more independent during optimization.

### 2.2 SYNAPSE

In neuroscience, synapses stand as pivotal junctions, orchestrating the complex symphony of neural communication. They serve as the critical interface between neurons, facilitating the transmission of information through chemical and electrical signals. In a Cyclic NN, we model the synapse $S$ as the edge between neurons defined in Section 2.1. Each synapse $S_{1,2} = (N_1 \to N_2)$ is a directional edge from computational neuron $N_1$ to $N_2$. It indicates the output of $N_1$, $\mathbf{h}_{\text{out}}^{N_1}$, will be propagated to $N_2$ as part of its input $\mathbf{h}_{\text{in}}^{N_2}$. Different from the current ANNs with DAG structure, the synapses between neurons can be organized as any connected graph structure, including the cyclic graph.

### 2.3 LOCAL OPTIMIZATION

Local optimization is the bedrock to support Cyclic NN, which is also a distinguishing point compared with current ANNs. For current ANNs, inputs are propagated through computational neurons to obtain the final representation. Designing a global loss function $\mathcal{L}_{\text{global}}$ based on the final representation and ground-truth labels is the *de facto* practice to train ANNs. $\mathcal{L}_{\text{global}}$ is optimized with BP algorithm to propagate the error signal layer-by-layer, which also prohibits the formation of computation cycle. Conversely, Cyclic NN depends on local optimization, *i.e.*, each computational neuron is optimized locally without gradients propagated from other computational neurons. For the computational neuron $N$, Cyclic NN has a local loss function $\mathcal{L}_{\text{Local}}$ to optimize its parameters. The local optimization principle is similar to the BI system, where each neuron can learn from its local context.

### 2.4 INFERENCE

During the inference phase, we design a readout layer to gather the output of all the neurons within the model as $\mathbf{h}_{\text{readout}} = f_{\text{readout}}([\mathbf{h}_{\text{out}}^{N_1}, \mathbf{h}_{\text{out}}^{N_2}, ..., \mathbf{h}_{\text{out}}^{N_{\mathcal{V}}}])$. $\mathbf{h}_{\text{readout}}$ collects all the encoded information and acts as the final

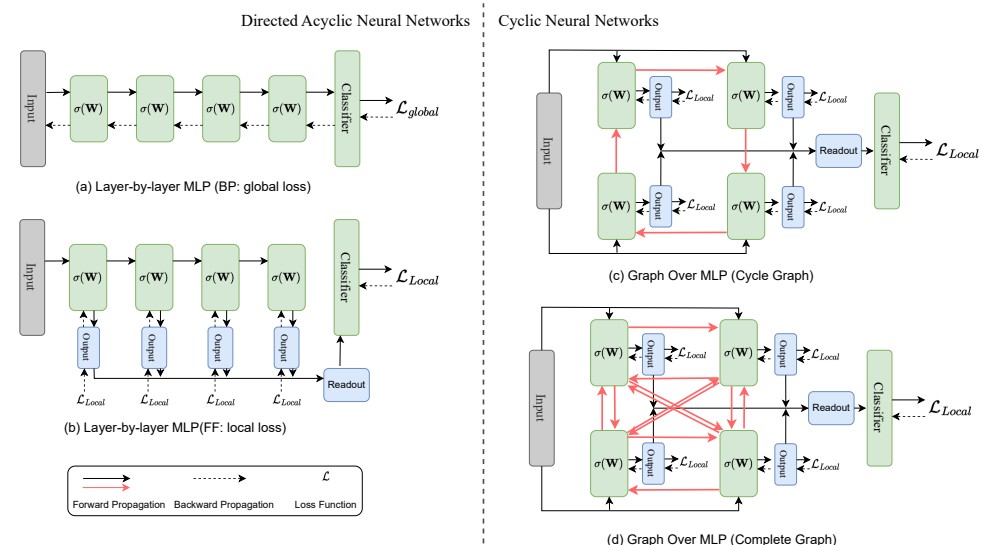

Figure 2: Comparison between different types of MLP structure.

representation for the inference task. For example, $\mathbf{h}_{\text{readout}}$ can be fed into a classifier for the classification task, where the classifier is trained together with computational neurons.

## 3 GRAPH OVER MULTI-LAYER PERCEPTRON

We propose the first Cyclic NN under the most generalized case, Graph Over Multi-Layer Perceptron (GOMLP), to show the design principle of Cyclic NN. As shown in Figure 2(c) and (d), GOMLP is designed by building a graph structure over the multi-layer perception to solve the classification task [1].

### 3.1 INPUT CONSTRUCTION

For the classification task, each sample is symbolized as the feature-label pair $(\mathbf{h}_i, y_i)$, where $\mathbf{h}_i$ is the representation of sample $i$ and $y_i$ is the corresponding label. To enable the local optimization illustrated in Section 3.4, a fusion function is used to construct the input as:

$$
\begin{aligned}
\mathbf{h}_{\text{pos}} &= f_{\text{fusion}}(\mathbf{h}, \mathbf{y}_{\text{true}}) = \mathbf{h}||\mathbf{y}_{\text{true}}, \\
\mathbf{h}_{\text{neg}} &= f_{\text{fusion}}(\mathbf{h}, \mathbf{y}_{\text{false}}) = \mathbf{h}||\mathbf{y}_{\text{false}}, \\
\mathbf{h}_{\text{neu}} &= f_{\text{fusion}}(\mathbf{h}, \mathbf{y}_{\text{neutral}}) = \mathbf{h}||\mathbf{y}_{\text{neu}},
\end{aligned}
\tag{1}
$$

$\mathbf{h}_{\text{pos}}$, $\mathbf{h}_{\text{neg}}$, and $\mathbf{h}_{\text{neu}}$ are the constructed input for local optimization of different parts. $f_{\text{fusion}}$ is a function to fuse information between feature and label, which is defined as a concat function ($||$). $\mathbf{y}_{\text{true}}$ is the one-hot vector of ground-true label, $\mathbf{y}_{\text{false}}$ is the one-hot vector of a randomly sampled false label. For $\mathbf{y}_{\text{neu}}$, we place an $\frac{1}{\text{Class Number}}$ on all the dimensions of one-hot vector to indicate $\mathbf{h}_{\text{neutral}}$ is neutral to all classes. $f_{\text{fusion}}$ can be designed as any proper function to fuse information of the input feature and the label. In our study, we design it as a simple concat function same as (Hinton, 2022).

### 3.2 COMPUTATION GRAPH

The computation graph $\mathcal{G}$ contains the computational neurons $\mathcal{V}$ and the synapses $\mathcal{E}$. Each computational neuron $N \in \mathcal{V}$ is a local module for calculation and optimization, while synapse $S$ defines how the information

---

[1] Code is released at https://anonymous.4open.science/r/Cyclic-Neural-Network-025F/README.md

propagates between computational neurons. $\mathcal{G}$ can be defined as a graph generator:

$$\mathcal{G} = \text{Generator}(|\mathcal{V}|, |\mathcal{E}|). \tag{2}$$

The above-generated graph $\mathcal{G}$ denotes a general graph structure. Meanwhile, to justify the effectiveness of the proposed Cyclic NN, we test multiple graph generators in this paper.

- Chain graph. Neurons are organized layer-by-layer as shown in Figure 2(b). In this case, GOMLP degrades to Hinton's method (Hinton, 2022).
- Cycle graph. Neurons form a cycle by connecting the neurons head-to-tail as shown in Figure 2(c).
- Complete graph. Each neuron connects to all the other neurons, as shown in Figure 2(d).
- Watts-Strogatz (WS) graph (Watts & Strogatz, 1998). It produces graphs with small-world properties, including short average path lengths and high clustering.
- Barabási–Albert (BA) graph (Albert & Barabási, 2002). It generates random scale-free networks using a preferential attachment mechanism.

### 3.2.1 NEURON UPDATE

In GOMLP, each neuron is parameterized by a linear layer. At each propagation, neuron $N$ is updated by (we omit the notation of $N$ in equation for simplicity):

$$\mathbf{h}_{\text{out}} = \sigma(\mathbf{W}\tilde{\mathbf{h}}_{\text{in}}), \tag{3}$$

where $\sigma$ is the Relu activation function (Nair & Hinton, 2010), $\mathbf{W} \in \mathbb{R}_{d_{\text{out}}^N \times d_{\text{in}}^N}$ is $N$'s parameter. $d_{\text{out}}^N$ is $N$'s output dimension, and $d_{\text{in}}^N$ is $N$'s input dimension, which is defined by the output of $N$'s pre-synapse neurons. $\tilde{\mathbf{h}}_{\text{in}}$ is the normalized input as $\tilde{\mathbf{h}}_{\text{in}} = \frac{\mathbf{h}_{\text{in}}}{\|\mathbf{h}_{\text{in}}\|_2}$, where $\mathbf{h}_{\text{in}}$ is computational neuron $N$'s input.

### 3.2.2 SYNAPSE PROPAGATION

Each synapse $S = (N_i \to N_j)$ is a directional edge from computational neuron $N_i$ to $N_j$, which indicates $N_i$ is the pre-synapse neuron of $N_j$ and $\mathbf{h}_{out}^{N_i}$ (the output of $N_i$) will be propagated to $N_j$. Assume for neuron $N$, we obtain a set of pre-synapse neurons $(N_1, N_2, ..., N_n)$ based on the topology of $\mathcal{G}$. Then, in each propagation, $N$ receives the output of all its pre-synapse neurons along the synapses and fuse the information to form its input by a concatenation function:

$$\mathbf{h}_{\text{in}} = \mathbf{h}||\mathbf{h}_{\text{out}}^{N_1}||\mathbf{h}_{\text{out}}^{N_2}||, ..., ||\mathbf{h}_{\text{out}}^{N_n}, \tag{4}$$

where $||$ is the concat function, $\mathbf{h}$ is the input representation constructed in Section 3.1. Then we can obtain $\mathbf{h}_{\text{in,pos}}$, $\mathbf{h}_{\text{in,neg}}$, $\mathbf{h}_{\text{in,neu}}$ by providing $\mathbf{h}_{\text{pos}}$, $\mathbf{h}_{\text{neg}}$, $\mathbf{h}_{\text{neu}}$ separately. As we relax the layer-by-layer restriction, the differentiation between the input/hidden/output layers is also relaxed. We directly put the input $\mathbf{h}$ to all computational neurons. Thus, the input dimension size of $N$, $d_{\text{in}}^N = d_{\mathbf{h}} + d_{\text{out}}^{N_1} + d_{\text{out}}^{N_2} + ... + d_{\text{out}}^{N_n}$.

## 3.3 READOUT LAYER

The readout layer collects information from all computational neurons. The input of the readout layer is the concat function of all computational neurons as $\mathbf{h}_{\text{in}}^{\text{readout}} = f_{\text{readout}}(\mathbf{h}_{\text{out}}^{N_*}) = ||_{i=1}^{|\mathcal{V}|}(\mathbf{h}_{\text{out}}^{N_i})$, where $||$ is the concat function. Then, the readout layer casts the representation to output dimension:

$$\hat{\mathbf{y}} = \text{Softmax}(\mathbf{W}_{\text{readout}}\mathbf{h}_{\text{in}}^{\text{readout}}), \tag{5}$$

where $\mathbf{W}_{\text{readout}} \in \mathbb{R}^{\text{Class Number} \times d(\mathbf{h}_{\text{in}}^{\text{readout}})}$ is the parameter of the readout layer and $\hat{\mathbf{y}}$ is the prediction vector.

### 3.4 LOCAL OPTIMIZATION

GOMLP comprises two parts that hold parameters: the computational neurons, and the readout layer. All parameters are optimized locally without gradient propagates between each parts. The computational neuron and readout layer are optimized differently with different inputs, which is shown in Algorithm 1, which contains the optimization of each computational neuron and the readout layer. The hyper-parameter $T$ indicates the number of synapse propagation over $\mathcal{G}$.

#### 3.4.1 COMPUTATIONAL NEURON OPTIMIZATION

Computational neurons are optimized to differentiate the positive examples from negative ones. For computational neuron $N$, its optimization involves $\mathbf{h}_{\text{in,pos}}$ and $\mathbf{h}_{\text{in,neg}}$. After the computational neuron update (Equation 3), we can get $\mathbf{h}_{\text{out,pos}}$ and $\mathbf{h}_{\text{out,neg}}$, respectively. Then, following (Hinton, 2022), a goodness function is used to calculate the goodness score as $p(\mathbf{h}) = \sigma(\sum_i h_i^2 - \theta * d(\mathbf{h}))$, where $p(\mathbf{h})$ is the goodness score of $\mathbf{h}$, $d(\mathbf{h})$ is the dimension size of $\mathbf{h}$, $\sigma$ is the Relu activation function and $\theta$ is the threshold hyper-parameter. The binary cross-entropy loss is used to optimize each computational neuron:

$$\mathcal{L}_N = -\frac{1}{|\mathcal{D}|}\sum_{\mathcal{D}}(\log(p(\mathbf{h}_{\text{out,pos}})) - \log(p(\mathbf{h}_{\text{out,neg}}))), \mathbf{W}_N \leftarrow \mathbf{W}_N - lr\nabla_{\mathbf{W}_N}\mathcal{L}_N, \quad (6)$$

where $\mathcal{D}$ is the dataset, $\mathbf{W}_N$ is the parameter of computation neuron $N$ and $lr$ is the learning rate. The optimization of computational neurons aims to increase the neuron's output for positive samples while decreasing the neurons' output for negative samples. It enables each computational neuron its own ability to differentiate positive examples from negative ones.

#### 3.4.2 READOUT LAYER OPTIMIZATION

The readout layer is designed to accomplish the classification task for GOMLP. It reads the information from all computational neurons, and makes the decision over classes. To relieve the label leakage issue, the readout layer is only optimized with $\mathbf{h}_{\text{neutral}}$. It is optimized by a multi-class cross-entropy loss:

$$\mathcal{L}_{\text{Readout}}(\mathbf{y}, \hat{\mathbf{y}}) = -\frac{1}{|\mathcal{D}|}\sum_{|\mathcal{D}|}\sum_{c=1}^{C} y_c \log(\hat{y}_c), \quad (7)$$

where $C$ is the number of classes, $\mathbf{y}$ is the one-hot vector of ground-truth label and $\hat{\mathbf{y}}$ is the prediction from Equation 5. Then the parameter within readout layer is optimized by $\mathbf{W}_{\text{readout}} \leftarrow \mathbf{W}_{\text{readout}} - lr\nabla_{\mathbf{W}_{\text{readout}}}\mathcal{L}_{\text{Readout}}$. Though the optimization of computational neuron and readout layer are localized and different, these two parts complement each other. Computational neurons aim to extract the hidden representations, and the readout layer aims to make the final decision.

---

**Algorithm 1** Cyclic NN Framework Optimization

**Input**: dataset $\mathcal{D}$
**Parameter**: $\mathcal{G} = (\mathcal{V}, \mathcal{E})$, $\mathbf{W}_{\text{readout}}$
**Output**: $\mathcal{G}$, $\mathbf{W}_{\text{readout}}$

1: **while** *Not Converged* **do**
2:     Obtain inputs by Eq. 1 from $\mathcal{D}$
3:     Let $t = 0$
4:     **while** $t < T$ **do**
5:        **for** $N \in \mathcal{V}$ **do**
6:           Synapse Propagate by Eq. 4
7:           Computational neuron Update by Eq. 3
8:           Optimize $N$ by Eq. 6
9:        **end for**
10:        $t = t + 1$
11:     **end while**
12:     Calculate the output of Readout layer by Eq. 5
13:     Optimize $\mathbf{W}_{\text{readout}}$ by Eq. 7
14: **end while**
15: **return** $\mathcal{G} = (\mathcal{V}, \mathcal{E})$, $\mathbf{W}_{\text{readout}}$

---

During the inference time, we pair each test sample with the neutral label to construct $\mathbf{h}_{\text{neu}}$. It then propagates through the GOMLP to obtain its representation on each computational neuron. Finally, we predict its class with the largest logit from the output of the readout layer.

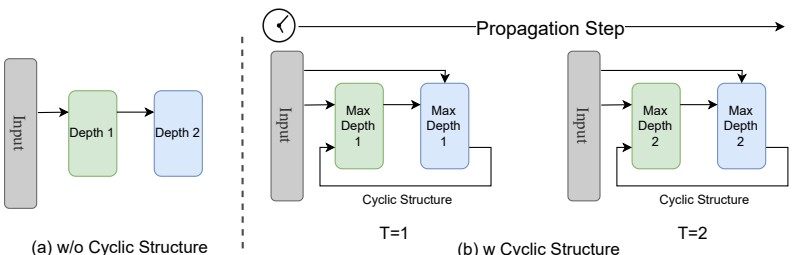

Figure 3: Cycic Structure increases model depth with $T$ with no extra parameter cost.

### 3.5 COMPLEXITY ANALYSIS

Assume we have a Cyclic NN represented as $\mathcal{G} = (\mathcal{V}, \mathcal{E})$, where each $v \in \mathcal{V}$ is a linear layer and each $e \in \mathcal{E}$ represents an edge between two linear layers. To make the analysis easier, let's firstly assume the time complexity for each computation neuron is $O(1)$. Then given timestep $T$, the input goes within the graph structure and cycles along all edges within $\mathcal{E}$ for $T$ times, which takes $O(T * |\mathcal{E}|)$. Within each cycle, a $O(|\mathcal{V}|)$ is needed to calculate both the forward/backward pass within each computation block. The total time complexity during each step training is $O(T * |\mathcal{E}| * |\mathcal{V}|)$. Inference time complexity is the same as we only omit the backward pass. By comparison, current layer-by-layer structure is a special case for Cyclic NN when the graph is organized as chain graph and enabling back-propagation between layers to remove $T$. Under this case, the time complexity of current layer-by-layer form would be $O(|\mathcal{V}|)$.

To be noted that, experiment in Section 4.3 finds the timestep $T$ is usually very small ($\leq 5$ in all tested datasets). So the complexity of proposed Cyclic NN can be $O(|\mathcal{E}| * |\mathcal{V}|)$ in practice. Besides, Cyclic NN is also optimized with local losses on each computation block, which frees the backward lock problem Dean et al. (2012); Löwe et al. (2019) inherited within the Backward-Propagation algorithm. Cyclic NN enables asynchronous parallel update of all computation neurons $\mathcal{V}$ at the same time. So the time complexity of Cyclic NN can be further reduced to $O(|\mathcal{E}|)$ in practice.

Let's take the example of GOMLP and further consider the time complexity of each computation neuron. The maximum complexity for each computation neuron is $O(|\mathcal{V} - 1|d * d) = O(|\mathcal{V}|d^2)$ when it receives information from all the other computation neurons. So the total time complexity of GOMLP is $O(|\mathcal{E}||\mathcal{V}|d^2)$.

### 3.6 ANALYSIS ON ADVANTAGE OF CYCLIC STRUCTURE

With piecewise linear function (such as ReLu Nair & Hinton (2010) used in this work) as the activation function, the neural network splits the whole input space as different linear regions, and the network expressiveness can be quantitatively measured by the maximal number of those regions Montufar et al. (2014); Raghu et al. (2017). Previous research has proved a rectifier neural network with $n_0$ input units and $L$ hidden layers of width $n \geq n_0$ can compute functions that have $\Omega((\frac{n}{n_0})^{(L-1)n_0} n^{n_0})$ linear regions Montufar et al. (2014). It shows neural network depth has an exponential advantage impact on its expressive power.

To analyze the impact of the proposed cyclic structure on the network's expressiveness, we compare two scenarios: one without cyclic connections and another with cyclic connections, as illustrated in Figure 3(a) and (b), respectively. In the absence of a cyclic structure, as shown in Figure 3(a), the network depth remains fixed, determined solely by the number of layers. However, when a cyclic structure is introduced, as depicted in Figure 3(b), the model depth effectively increases with the propagation steps $T$. Specifically, at $T = 1$, the output of each layer corresponds to a depth of 1, as it directly incorporates information from the input. At $T = 2$, each layer aggregates two types of information: depth-0 information directly from the input and depth-1 information propagated from neighboring computational neurons, resulting in a maximum depth of 2. As $T$ increases, the depth of the information available to each layer grows proportionally, enhancing the network's expressiveness. The cyclic structure increases the model's effective depth through iterative propagation, allowing the network to achieve greater expressiveness without additional parameters.

Table 1: Error rate (%) ↓ on different datasets.

| Train | Graph | MNIST | NewsGroup | IMDB |
|---|---|---|---|---|
| MLP-Ensemble | - | $1.91_{\pm 0.21}$ | $45.35_{\pm 0.84}$ | $17.36_{\pm 0.23}$ |
| BP | Chain* | $1.77_{\pm 0.16}$ | $42.11_{\pm 0.92}$ | $\mathbf{17.16}_{\pm 0.19}$ |
| FF | Chain | $1.83_{\pm 0.2}$ | $43.88_{\pm 0.28}$ | $18.75_{\pm 0.92}$ |
| BP | Chain | $1.74_{\pm 0.11}$ | $38.85_{\pm 0.42}$ | $17.27_{\pm 0.13}$ |
| FF | Cycle | $1.80_{\pm 0.14}$ | $43.54_{\pm 0.41}$ | $18.97_{\pm 0.49}$ |
| FF | WSGraph | $1.70_{\pm 0.17}$ | $38.28_{\pm 0.13}$ | $17.93_{\pm 0.28}$ |
| FF | BAGraph | $1.64_{\pm 0.08}$ | $38.41_{\pm 0.14}$ | $18.20_{\pm 0.67}$ |
| FF | Complete | $\mathbf{1.54}_{\pm 0.05}$ | $\mathbf{38.266}_{\pm 0.06}$ | $17.58_{\pm 0.20}$ |

## 4 EXPERIMENTS

### 4.1 BASELINES

We compared GOMLP with a variant of different methods to reveal the advantages of graph-structured MLP, which can be differentiated by two attributes (Training and Graph). Training indicates the training method, where BP indicates Backward Propagation (Rumelhart et al., 1986) and FF indicates the Forward-forward algorithm (Hinton, 2022). The graph indicates the graph structure of computational neurons. We keep 4 computation neurons for all methods during the experiments. The special cases are further illustrated as:

• BP-Chain*: Layer-by-layer networks trained with BP as depicted in Figure 2(a). It is the current default way of building and training ANNs.

• FF-Chain: Layer-by-layer networks trained with FF as depicted in Figure 2(b) same as (Hinton, 2022).

• BP-Chain: A modified version of BP-Chain*, where we use the structure of Figure 2(b) and trained with BP. It adds direct local supervision on each layer.

FF-Cycle, FF-WSGraph, FF-BAGraph, and FF-Complete are different versions of GOMLP, where the training is FF and only the graph generator defined in Eq. 2 differs.

### 4.2 OVERALL COMPARISON

The overall experiment result is shown in Table 1. We can have several interesting and exciting findings:

• FF-Complete achieves the best performance on MNIST and NewsGroup datasets and comparable results to the best one on the IMDB dataset. It is the first FF-trained model that outcompetes the BP-trained model. It is an exciting observation of the effectiveness of the FF algorithm compared with the BP algorithm.

• FF-Chain performs worse than BP-Chain* on all datasets. This observation is on par with (Hinton, 2022), where the FF lags behind the BP training algorithm when they both follow layer-by-layer organization as a chain graph. However, we can surpass BP-Chain* when organizing the computational neurons as a graph structure. This finding inevitably reveals the advantages of GOMLP by organizing multi-layer perceptron as a flexible graph structure.

• FF-Cycle achieves similar performance with FF-Chain on three datasets. It is reasonable because these two methods have only one edge difference. When we build more complex graphs (WSGraph, BAGraph, Complete Graph), we can observe much better performance immediately. It shows the benefits of enriching the communication between computational neurons by the GOMLP.

• BP-Chain is better than BP-Chain* in most cases. Compared with BP-Chain*, BP-Chain further adds layer-wise optimization directly from the final loss. It indicates the advantageous layer-wise optimization, which provides new guidelines when designing layer-by-layer neural networks.

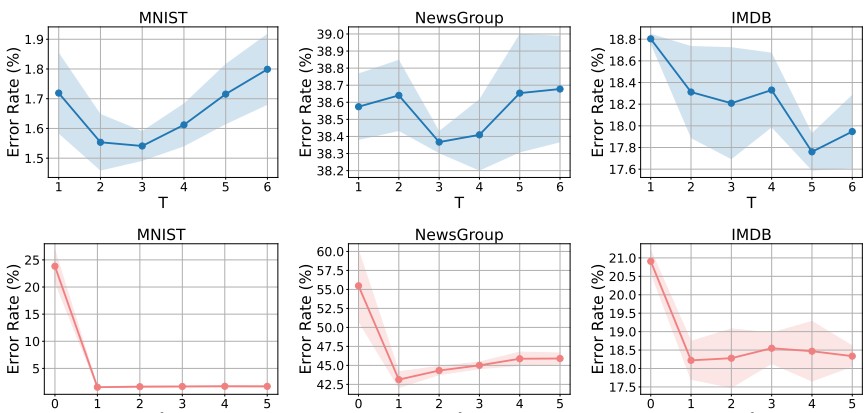

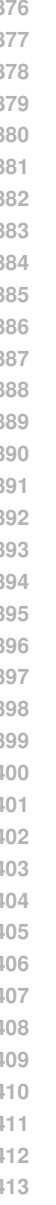

Figure 4: Parameter sensitivity of $T$ and $\theta$

In summary, the experiment results answer that we do not need to stack neural networks layer-by-layer sequentially, and we can organize the neural networks as a flexible, complex graph structure like the brain. More excitingly, we can outperform the current *de facto* layer-by-layer neural network design paradigm with the Cyclic NN, and provide a totally new way of building ANNs.

### 4.3 HYPER-PARAMETER SENSITIVITY

This section tests the impact of introduced hyper-parameters ($T$ and $\theta$). Results are shown in Figure 4. $T$ controls the number of propagation between computational neurons. Larger $T$ indicates more times the information is propagated. We can observe an error rate trend that first decreases and then increases on all three datasets. It indicates that the computational neurons need a suitable propagation number. When $T$ is small, computational neurons can not draw sufficient lessons from each other. When $T$ is large, computational neurons are over-propagated, which leads to the over-smoothing problem. Different from graph neural network (Chen et al., 2020) where the graph is data and the over-smoothing occurs among node representation, in GOMLP, the graph is the model, and the over-smoothing occurs among computational neurons.

$\theta$ controls the goodness threshold of each computational neuron. We can observe a sharp error rate decrease when $\theta$ increases from 0 to 1, and then it gets stable with larger $\theta$. It indicates the existence of the goodness threshold matters more than the threshold value. When $\theta = 0$, there is little room to optimize the computational neuron, which can lead to the training collapse as the computational neuron can not differentiate the negative samples. When $\theta$ is larger, there is more room to optimize the goodness score toward the negative samples, as all the goodness scores under the threshold can represent a negative sample.

### 4.4 ABLATION STUDY

This section studies the impact of different optimization modules within GOMLP, including the computational neuron optimization $\mathcal{L}_N$ and readout layer optimization $\mathcal{L}_{\text{Readout}}$. We conduct experiments on the FF-Complete structure, and the results are sum-

Table 2: Error rate (%) ↓ of Ablation study.

| Model | MNIST | NewsGroup | IMDB |
|---|---|---|---|
| FF-Complete | **1.54** | **38.26** | **18.20** |
| -$\mathcal{L}_N$ | 2.24 | 47.61 | 22.94 |
| -$\mathcal{L}_{\text{Readout}}$ | 95.58 | 95.55 | 44.26 |

marized in Table 2. We can have the following observations: 1) The error rate increases when removing any optimization module, indicating the usefulness of each component. 2) GOMLP falls to a very large error rate (nearly random guess) when removing $\mathcal{L}_{\text{Readout}}$. It is reasonable as we depend on the readout layer to complete the final classification task. Without optimization on the readout layer, GOMLP falls into random guess even with optimized computational neuron's input. 3) The error rate increases by removing $\mathcal{L}_N$. It shows the

computational neuron's optimization can provide a more informative goodness score for the readout layer to complete the classification task. $\mathcal{L}_N$ and $\mathcal{L}_{\text{Readout}}$ complement each other within GOMLP.

## 5 RELATED WORK

### 5.1 ARTIFICIAL NEURAL NETWORKS

Artificial neural networks (ANNs) have evolved through various paradigms, each suited to specific tasks and data structures. Feedforward neural networks (MLPs Rumelhart et al. (1986), CNN LeCun et al. (1995), and Transformers Vaswani et al. (2017), etc) form the foundational class of ANNs. These models are characterized by their layer-by-layer processing, making them effective for structured data tasks He et al. (2016); Vaswani et al. (2017). Recurrent neural networks (RNNs) Elman (1990) and their variants such as Long Short-Term Memory Schmidhuber et al. (1997) and Gated Recurrent Units Cho (2014) introduced recurrent loops, enabling temporal modeling for sequential data. Graph Neural Networks (Graph Convolutional Networks Kipf & Welling (2016), Graph Attention Networks Velickovic et al. (2017), and Graph Isomorphism Networks Xu et al. (2018), etc) extend neural computation to graph-structured data. While GNNs support message passing between nodes, they are typically constrained by acyclic computational graphs. Recently, there are also new ANN designs inspired by biology nerve systems. Liquid neural networks Hasani et al. (2021) adapt dynamically to changing inputs, exhibiting flexible, real-time computation inspired by biological intelligence. Spiking Neural Networks Tavanaei et al. (2019) mimics the communication pattern of biology neurons with discrete spike events instead of continuous activations.

Cyclic NN firstly focuses on the network topology similarity with biology neural network by introducing cyclic structures within ANNs. It represents a transformative departure from these existing paradigms by removing the Directed Acyclic Graph (DAG) constraint. Inspired by the flexible and dynamic nature of biological neural systems, Cyclic NN introduces cyclic connections between neurons, enabling richer information flow. This design achieves enhanced expressiveness, biological plausibility, and flexibility.

### 5.2 LOCALIZED LEARNING ALGORITHM

Although the end-to-end BP algorithm is the dominant training algorithm for deep neural networks, studies have revealed notable limitations in such end-to-end training with global objectives (Bengio et al., 2015; Crick, 1989). Numerous works have proposed alternative training methods to make ANNs more biologically plausible. Inspired by Hebbian theory (Hebb, 2005), Hebbian Learning (Gerstner & Kistler, 2002) updates weights locally between two active, connected neurons, ensuring long-term stability so previously learned information is not lost. Addressing the weight transportation problem, Feedback alignment methods (Lillicrap et al., 2016; Nøkland, 2016) replace downstream synaptic weights with random weights, eliminating the need for feedback weights in neurons. Unlike the two phases of the BP algorithm, Equilibrium Propagation (Scellier & Bengio, 2017) performs both inference and weight updates using only one type of computation. The approach in (Nøkland & Eidnes, 2019) reduces memory consumption and increases training parallelism by adopting subnetworks and layer-wise training. (Hinton, 2022) introduces a simple yet efficient local objective function that measures the goodness of positive and negative data to optimize ANNs locally.

The localized learning algorithm is the bedrock that supports the cyclic structure within the neural networks. In this paper firstly beats global BP training with pure localized learning algorithm based on cyclic structure.

## 6 CONCLUSION

This research introduces Cyclic NN, a novel ANN architecture inspired by the complex, graph-like neural networks in biological intelligence. This transformative design diverges from traditional directed acyclic ANN structures. Our findings, demonstrated through the GOMLP model and validated on various datasets, showed enhanced performance over conventional DAG networks. This significant development paves the way for more flexible and biologically realistic AI systems, representing a major shift in ANN design.

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

## A APPENDIX

### A.1 DATASET

Table 3: Dataset Statistics

| Dataset | MNIST | NewsGroup | IMDB |
|---|---|---|---|
| # Training Samples | 50,000 | 9,314 | 20,000 |
| # Validation Samples | 10,000 | 2,000 | 5,000 |
| # Test Samples | 10,000 | 7,532 | 25,000 |
| # Dimensions | 784 | 788 | 770 |
| # Classes | 10 | 20 | 2 |

We conduct experiments on three widely studied datasets from computer vision and natural language processing domains. Data statistics are shown in Table 3. For each dataset, the training and test split follows the original setting. We further extract 20% samples from the training data as validation sets to tune hyper-parameters.

- MNIST [2] (LeCun et al., 1989). It contains handwritten digits from 0-9, which is the most accessible and used datasets in the field of machine learning.
- NewsGroup [3] (Lang, 1995). It is a collection of approximately 20,000 newsgroup documents, partitioned across 20 different newsgroups. This dataset is widely used for experiments in text applications of machine learning techniques, such as text classification and text clustering.
- IMDB [4] (Maas et al., 2011). It is a movie review dataset crawled from IMDB. It is the most widely studied dataset for binary sentiment classification.

For MNIST, we directly use its flattened pixel values as the input of all methods and replace the first 10 pixels with labels as the fusion function, which is the same as (Hinton, 2022) and leads to an input dimension of $28 * 28 = 784$. For NLP datasets (NewsGroup, IMDB), we use BERT (Devlin et al., 2018) to encode the sentences into a fixed-length tensor (768) as the input. The fusion function is the concat function, which leads to an input dimension of $768 + 20 = 788$ for NewsGroup and $768 + 2 = 770$ for IMDB dataset.

### A.2 EXPERIMENTAL SETTING.

We use Adam (Kingma & Ba, 2014) optimizer to train the model until it converges. Learning rate and weight decay are tuned within (0.1,0.01,0.001) and (0.0, 1e-2, 1e-4, 1e-6, 1e-8), respectively. The early stop technique is applied to avoid overfitting, where we stop training if there is no improvement on the validation set for continuous 10 epochs. We report the mean and variance on 20 experiments with different random seeds. All experiments are conducted on GeForce 4090 GPU.

### A.3 ADVANTAGES AND LIMITATIONS

Optimized with local losses, Cyclic NN is a novel ANN design paradigm that goes beyond DAG constraint. It has several advantages. (1) Flexibility. Cyclic NN can build and optimize neural networks in any graph structure. It provides more flexibility when designing ANNs. (2) Denser information flow. Cyclic NN supports

---

[2] http://yann.lecun.com/exdb/mnist/

[3] http://qwone.com/ jason/20Newsgroups/

[4] https://ai.stanford.edu/ amaas/data/sentiment/

direct denser information flow during the forward pass with the synapse construction among computation neurons. (3) Biology similarity. As discussed in Section 1, Cyclic NN is more biology-similar to the complex graph-structured networks, paving the way to minimize our understanding gap between artificial neural networks and biology neural networks.

Cyclic NN has several limitations. Its flexibility in building graph-structured networks leads to more network complexity. Cycles within the network also pose more challenges to the interpretability. Currently, there are no suitable training frameworks specifically designed for localized optimization, which hinders the optimization speed of Cyclic NN. This limitation can be solved with the development of localized training frameworks.

## A.4 NEURAL NETWORK COMPARISON BETWEEN RNN, GNN AND CYCLIC NN

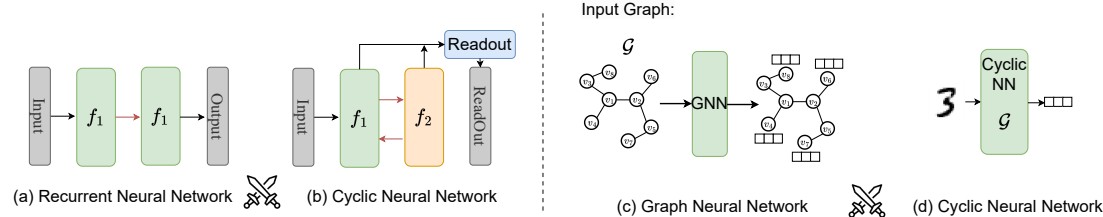

Figure 5: Neural Network Comparison between RNN, GNN and our proposed Cyclic NN.

This section makes a comparison among RNN, GNN and our newly proposed Cyclic NN. Figure 5(a) and (b) compares RNN and Cyclic NN. Recurrent structures (such as RNN, LSTM, GRU) focus on the recurrence of the same computation block. Our newly proposed Cyclic NN focus on the cyclic communication between different computation blocks as highlighted in red. The existence of cyclic structure enables building neural networks in any graph structure beyond DAG. Actually, recurrent structures can be seen as the self-loop over one computation neuron, and Cyclic NN enables much more flexible network structures beyond self-loop.

Figure 5(c) and (d) compare GNN and Cyclic NN. In GNNs (such as GCN, recurrent GNN, GAT), graph $\mathcal{G}$ is the input of network and GNN aims to learn node representation for each node. We usually use DAG structured computation within the model such as the linear layer in GCN. GNNs are the encoder of nodes within graphs, and the graph structure acts as the model's input. However, in Cyclic NN, input is not constrained to graphs. As shown in Figure 5(d), the input is one image and the Cyclic NN encodes inputs into representation. The graph structure $\mathcal{G}$ refers to the encoder itself within Cyclic NN.

## A.5 TRAINING CURVE OF CYCLIC NN UNDER DIFFERENT GRAPH STRUCTURES

The training curve of Cyclic NN on different graph structures are shown in Figure 6. In Cyclic NN, the optimization occurs locally at each computation neuron and the final classifier. FF Loss is the average forward-forward loss over all computation neurons. FF loss, classifier loss and error rate changes with training epochs are shown in the figure. We can observe that for all graph structures and datasets, the decrease of losses and error rate is very stable and steady. Localized optimization focuses on optimizing parameters at a local level without propagating updates across layers. This approach helps mitigate the gradient vanishing or exploding issues commonly encountered in global optimization.

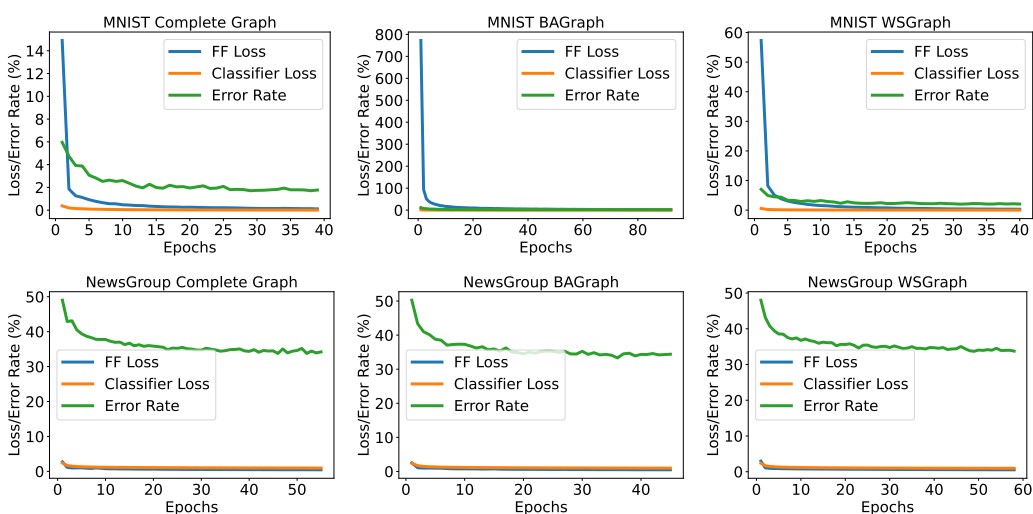

Figure 6: Training Curve of Cyclic NN under Different Graph Structure

