# OpenReview forum: "Beyond Directed Acyclic Computation Graph with Cyclic Neural Network"
_ICLR.cc/2025/Conference — Submitted to ICLR 2025_

### Official Review · Reviewer_ZyCj · 2024-10-29

**Soundness:** 3
**Presentation:** 2
**Contribution:** 2
**Rating:** 6
**Confidence:** 4

**Summary:**

The authors propose a biologically plausible neural architecture referred to as the cyclic NN. The defining characteristics of the cyclic NN are: (1) each neuron is parametrised as a linear layer, i.e. N to M rather than N to 1 mapping, (2) each (computational) neuron is trained locally, using the forward-forward algorithm (backpropagation across layers does not take place), (3) neuronal information is accumulated by a parameterised “readout” layer to allow for downstream tasks such as classification. As a result of the proposed architecture, the connections can be cyclic - i.e., directed acyclic graph (DAG) structure typical of NNs is not enforced.

**Strengths:**

**Originality:** The paper proposes a graph structure over MLPs, which is simple yet effective. The proposed architecture elegantly elevates layer-based neural models to a structure that inherently includes both recurrency and ensembling. As such, a higher degree of biological plausibility is achieved.

**Quality and clarity:** The paper is concise and clear, with very minor typos and grammatical mistakes. The ideas are elegant and simple, with potentially significant impact on the field. The code is made available.

**Significance:** The shift from DAGs to recurrent architectures is imminent, as such the paper is quite timely. The significance is diminished by the fact that the authors do not acknowledge any of the work on recurrent neural networks in their study. If the proposed model can be properly contextualised, I would be willing to accept the proposed method as more significant.

**Weaknesses:**

The closest existing NN architecture that is not a DAG is a recurrent NN (RNN). Plethora of research exists on RNNs, yet the authors do not mention this paradigm in the paper. How are the cyclic NNs different from RNNs? A critical discussion of this point is necessary. Similarly, a more expressive neuron can be compared to a memory block of a long short-term memory (LSTM) network. How does the proposed computational neuron differ from a gated neuron? Section 3.6 discusses how the cycle can be unrolled and interpreted as an arbitrary depth - which is exactly the argument for the adoption of recurrent architectures. The similarity is quite striking and cannot be ignored.

The authors compare their proposed cyclic NNs to traditional DAG architectures. I think a comparison to other biologically plausible architectures would be more applicable, e.g. liquid neural networks. Where do the cyclic NNs fit in the context of existing biologically plausible NNs? Section 5.1 briefly lists existing localised training algorithms, but does not properly put the proposed method in the context.

Another lacking comparison to existing methods is that to ensembling. Each neuron in the cyclic NN is essentially a one-layer MLP. Each MLP learns to differentiate between patterns. Then, the decisions of multiple MLPs are accumulated by the readout layer to make the final prediction. Isn’t this a form of weighted ensembling of MLPs?

**Questions:**

How does the proposed method differ from the multitude of recurrent architectures?

It is not clear how the parameters of the computational neurons and the readout layer are optimised. Is gradient descent employed? Please explain and/or provide the update equation for the weights.

Table 1 lists standard deviations. How many runs were used per each setup?

Page 2, line 79: “without waiting gradients” -> “without waiting for gradients”

Page 7, line 305: “Theme 4” - do you mean Section?

Page 7, line 318: “there is a neural network consists…” - grammatically incorrect, please re-write.

Page 7, line 324: origional -> original (typo)

---

> ### Author Response · Authors · 2024-11-20
> **Rebuttal on W1, W2 and Q1**
>
> We would like to express our sincere thanks for the reviewer’s constructive feedback. Based on your review, we have made corresponding changes within the paper and our rebuttals are provided below:
>
> **Rebuttal to Weakness 1 and Question 1:**
>
> *Concern:*
>
> Clarity between RNN and Cyclic NN.
>
> *Response:*
>
> To highlight the distinctive characteristics of Cyclic NN, we have added a new Section A.4 in the appendix, along with Figure 5, which provides a clear illustration comparing Cyclic NN with RNNs.
> Comparison with RNNs: Recurrent structures like RNNs, LSTMs, and GRUs focus on the recurrence of the same computation block. In contrast, our Cyclic NN emphasizes cyclic communication between different computation blocks, as highlighted in red in Figure 5. The presence of cyclic structures enables us to build neural networks with any graph structure beyond directed acyclic graphs (DAGs). While recurrent structures can be seen as self-loops over a single computation neuron, Cyclic NN allows for more flexible network structures beyond self-loops.
> We propose the computational neuron to increase the capacity of each calculation block. As a more  expressive neuron design, we can also use LSTM or gated neuron to be the computational neuron in Cyclic NN.
>
>
> **Rebuttal to Weakness 2:**
>
> To better contextualize our proposed Cyclic NN within current literature, we replace the related work of “Graph Generator” with “Artificial Neural Networks”. It can better contextualize the position of Cyclic NN. The newly added related work section is also given as follows:
>
> “Artificial neural networks (ANNs) have evolved through various paradigms, each suited to specific tasks and data structures. Feedforward neural networks (MLPs, CNN, and Transformers, etc) form the foundational class of ANNs. These models are characterized by their layer-by-layer processing, making them effective for structured data tasks. Recurrent neural networks (RNNs) and their variants such as Long Short-Term Memory and Gated Recurrent Units introduced recurrent loops, enabling temporal modeling for sequential data. Graph Neural Networks (Graph Convolutional Networks, Graph Attention Networks, and Graph Isomorphism Networks, etc) extend neural computation to graph-structured data. While GNNs support message passing between nodes, they are typically constrained by acyclic computational graphs.
> Recently, there are also new ANN designs inspired by biology nerve systems. Liquid neural networks adapt dynamically to changing inputs, exhibiting flexible, real-time computation inspired by biological intelligence. Spiking Neural Networks mimics the communication pattern of biology neurons with discrete spike events instead of continuous activations.
>
> Cyclic NN firstly focuses on the network topology similarity with biology neural network by introducing cyclic structures within ANNs. It represents a transformative departure from these existing paradigms by removing the Directed Acyclic Graph (DAG) constraint. Inspired by the flexible and dynamic nature of biological neural systems, Cyclic NN introduces cyclic connections between neurons, enabling richer information flow. This design achieves enhanced expressiveness, biological plausibility, and flexibility.”

---

> > ### Author Response · Authors · 2024-11-20
> > **Rebuttal on W3, Q2 and Q3**
> >
> > **Rebuttal to Weakness 3:**
> >
> > To answer this question, we conduct ensembling experiments of MLPs as another baseline. Experiment results are listed as follow:
> >
> > | Train        | Graph     | MNIST          | NewsGroup        | IMDB           |
> > |--------------|-----------|----------------|------------------|----------------|
> > | MLP-Ensemble | -         | 1.91±0.21      | 45.35±0.84       | 17.36±0.23     |
> > | BP           | Chain*    | 1.77±0.16      | 42.11±0.92       | **17.16**±0.19 |
> > | FF           | Chain     | 1.83±0.20      | 43.88±0.28       | 18.75±0.92     |
> > | BP           | Chain     | 1.74±0.11      | 38.85±0.42       | 17.27±0.13     |
> > | FF           | Cycle     | 1.80±0.14      | 43.54±0.41       | 18.97±0.49     |
> > | FF           | WSGraph   | 1.70±0.17      | 38.28±0.13       | 17.93±0.28     |
> > | FF           | BAGraph   | 1.64±0.08      | 38.41±0.14       | 18.20±0.67     |
> > | FF           | Complete  | **1.54**±0.05  | **38.266**±0.06  | 17.58±0.20     |
> >
> > We can observe that the MLP-Ensemble does not perform well on either datasets. Though the readout layer in Cyclic NN can be viewed as ensembling information of multiple MLPs, the core design of Cyclic NN is enabling the cyclic structures among MLPs, which distinguishes it from the ensembling methods. As we observed in the table, ensemble MLPs alone do not produce good results. But we can obtain the best performance by building cyclic structures among MLPs and ensemble the information with the readout layer. It also validates the importance of proposed cyclic structures.
> >
> >
> > **Rebuttal to Question 2:**
> >
> > All parameters within computational neurons and the readout layer are optimized using gradient descent. Based on your suggestion, we have added the update equation in Section 3.4.1 and Section 3.4.2 within our paper to make this point clearer.
> >
> >
> > **Rebuttal to Question 3:**
> >
> > As stated in Appendix A.2,  we report the mean and standard deviations on 20 experiments with different random seeds for all experiments.
> >
> > Besides, we also make corresponding changes and proofread the paper again to eliminate typos. We would like to express our thanks for your careful review and detailed feedback. Let us know if there are any other concerns regarding Cyclic NN :).

---

> > > ### Comment · Reviewer_ZyCj · 2024-11-26
> > > **Thank you for the additional experiments**
> > >
> > > I appreciate the comparison to ensembling MLPs. I still feel that the link to RNNs should come out stronger and earlier in the main body of the paper rather than being pushed out to the appendices.

---

### Official Review · Reviewer_5dgC · 2024-11-01

**Soundness:** 3
**Presentation:** 2
**Contribution:** 3
**Rating:** 6
**Confidence:** 4

**Summary:**

This paper proposes a new design of artificial neural networks. The novelty lies in the fact that they don't have a directed acyclic graph (DAG) structure. This is a fundamental innovation because the training of neural networks nowadays depends on the DAG structure so that the gradient of the global loss function can be computed. To support the new architecture, the authors follow the forward-forward algorithm proposed by Hinton (2022), where local losses are used to train individual neurons and the final classifier. The authors demonstrate experiment results, which for the first time suggest that forward-forward training can outperform standard back-propagation training.

**Strengths:**

- This paper proposes a ground-breaking, innovative idea to build neural networks.

- The idea is backed by attractive experiment results.

- The proposed neural network, Cyclic NN, is a step forward toward a drastically different paradigm of machine learning models that are more biologically sound.

**Weaknesses:**

Some technical details are unclear. See the following "Questions" section.

Additionally, it would be informative to experimentally compare the proposed architecture with GNNs due to their similarities. Every node in the GNN takes the same input feature and the GNN uses a readout layer similar to the readout in Cyclic NN. In this case, the main difference between a GNN and a Cylic NN is that the GNN uses the same $W$ matrix for every node in a layer and uses different $W$ matrices for different layers, while the Cyclic NN uses different $W$ matrices for different nodes. In this regard, GNN is more parameter efficient. Of course, the training method is fundamentally different. Which architecture performs better?

**Questions:**

The main question surrounds Eqn (4), which causes confusion when the reader tries to connect it with the inner while loop of Algorithm 1, Figure 2(b) as a special case, and the discussions about unrolling in Section 3.6.

In Eqn (4), the neuron input depends on the outputs of the adjacent neurons. When $t = 0$, the outputs of the adjacent neurons have not been unknown yet. So how is line 6 of Algorithm 1 computed?

Do the authors ignore the outputs of the adjacent neurons when $t = 0$?

If so, we further look into the for loop of Algorithm 1. This loop loops over the neurons $N$. For a later $N$, if it is adjacent to an earlier neuron (call it $N_1$), would the input of the later neuron use the output of the earlier neuron in the last round (before the for loop) or in the current round (inside the for loop)?

In either answer to the above question, it appears that every neuron has one parameter matrix $W$ (as opposed to a few). The inner while loop of Algorithm 1 updates this parameter $T$ times. Is this understanding correct?

If correct, then the $T$ steps of the inner while loop of Algorithm 1 do not propagate information across $T$ hops of the graph. Rather, information is propagated to at most one hop away, no matter how big is $T$. The inner while loop is more like running an optimization $T$ steps rather than propagating information in a $T$-layer GNN.

If the above understanding is correct, then the unrolling in Figure 3 does not make sense. Consider the two $\sigma(W_1)$ on the right of Figure 3. They are the same neuron at different training stages. The first $\sigma(W_1)$ takes the value obtained after line 8 of Algorithm 1, when $t=0$. The second $\sigma(W_1)$ takes the value at $t=1$. This is very different from unrolling an RNN, where the RNN cell uses the same parameter values at different times.

If the above understanding is correct, then the discussion of the expressive power in Section 3.6 is dubious, because the neural network does not have a depth $T$ like in a usual neural network.

Now get back to Eqn (4). The neuron input includes the input representation $h$. This does not seem to be the case in Figure 2(b). If one considers the architecture in Figure 2(b) a special case of Cyclic NN, then following the convention in Figure 2(c) and (d), the black arrows that chain the neurons should be red arrows instead. Moreover, the input should have a black arrow pointing to every neuron.

Do the authors really mean Figure 2(b) to be in the current form, or in the edited form elaborated above? For FF-Chain, do the authors mean the current Figure 2(b) or the edited form? What about BP-Chain and BP-Chain*?

---

> ### Author Response · Authors · 2024-11-20
> **Rebuttal to Weakness**
>
> We would like to firstly thank the reviewer for acknowledging our novelty, contribution and foreseeing the future impact of our proposed Cyclic NN. Our rebuttals towards your weaknesses and questions are listed as follows. Hope we are able to clear your concerns on this paper.
>
> **Rebuttal to Weakness:**
>
> *Concern:*
>
> Differences between Cyclic NN and Graph Neural Network is not clear
>
> *Response:*
>
> To highlight the distinctive characteristics between Cyclic NN and graph neural networks.  We specifically add a new Section A.4 in the appendix, along with Figure 5, which provides a clear illustration comparing Cyclic NN and GNN.
>
> Comparison with GNNs: In GNNs (such as GCNs, recurrent GNNs, and GATs), the graph $\mathcal{G}$ is the input to the network, aiming to learn representations for each node. Typically, DAG-structured computations are used within the model, like the linear layers in GCNs. GNNs serve as encoders for nodes within graphs, with the graph structure acting as the model's input. However, in Cyclic NN, the input is not constrained to graphs; it can be an image, for example, and the Cyclic NN encodes this input into a representation. Here, the graph structure $\mathcal{G}$ refers to the encoder itself within the Cyclic NN. Thus, the Cyclic NN has fundamental differences with GNNs.

---

> > ### Comment · Reviewer_5dgC · 2024-11-22
> > **Rebuttal read**
> >
> > The rebuttal answers my questions. I am happy with it.

---

> ### Author Response · Authors · 2024-11-20
> **Rebuttal to Q1-4**
>
> **Rebuttal to Question:**
>
> We would like to provide the rebuttal based on the list of your questions.
>
> **Q1:** In Eqn (4), the neuron input depends on the outputs of the adjacent neurons. When t=0, the outputs of the adjacent neurons have not been unknown yet. So how is line 6 of Algorithm 1 computed? Do the authors ignore the outputs of the adjacent neurons when t=0?
>
> **A1:** We initialize all computational neuron’s output to 0 when t=0. Thus, we pass the tensor with 0s on line 6 of Algorithm when t=0, which ignores the outputs of adjacent neurons.
>
>
>
>
> **Q2:** For a later $N$, if it is adjacent to an earlier neuron (call it $N_1$), would the input of the later neuron use the output of the earlier neuron in the last round (before the for loop) or in the current round (inside the for loop)?
>
> **A2:** This is a very good question. We also face this problem when designing the training algorithm. We finally adopted the output of the earlier neuron in the last round (before the for loop) for propagation as we found the training would be more stable compared to the other choice. If we use the current round result, the computational neurons update will depend on the update order, which we also want to avoid as there is no reason to pre-define the update order especially on the cyclic graph structure.
>
>
>
>
> **Q3:** The inner while loop of Algorithm 1 updates this parameter $T$ times. Is this understanding correct?
>
> **A3:** Yes, this understanding is correct. We will update each computational neuron $T$ times.
>
>
>
>
> **Q4:** If correct, then the $T$ steps of the inner while loop of Algorithm 1 do not propagate information across $T$ hops of the graph. Rather, information is propagated to at most one hop away, no matter how big is $T$. The inner while loop is more like running an optimization $T$ steps rather than propagating information in a $T$-layer GNN.
>
> **A4:** Here we would like to argue the information will be propagated across $T$ hops of the graph similar to the message passing mechanism in GNNs. In each step, the computational neuron receives information from its neighbors and gets the new output for next step’s propagation. Here its parameter only updates by one training step, which makes it stay nearly the same. Its output will not change and still keeps its neighbors information. In the next round, its output (carrying current round neighbor’s information) will be propagated again to reach farther neighbors. Here, we optimize $T$ steps, and at the same time, the information is propagated to $T$ hops neighborhoods.
>
> It also answers the following question that as the information is propagated further, the unrolling in Figure 3 is reasonable because the output is propagated further and splitted into more linear regions by Relu activation.

---

> > ### Author Response · Authors · 2024-11-20
> > **Rebuttal to Q5 and Q6**
> >
> > **Q5:** If the above understanding is correct, then the discussion of the expressive power in Section 3.6 is dubious, because the neural network does not have a depth $T$ like in a usual neural network.
> >
> > **A5:** We would like to thank the reviewer for pointing out the dubious analysis in Section 3.6. After a careful consideration, we replace the analysis in Section 3.6 to better match our model. We also provide a new Figure 3 to help understand the analysis. Our revised analysis is provided as follows (It is better to read together with the newly replaced Figure 3):
> >
> > “To analyze the impact of the proposed cyclic structure on the network's expressiveness, we compare two scenarios: one without cyclic connections and another with cyclic connections, as illustrated in Figure3(a) and (b), respectively. In the absence of a cyclic structure, as shown in Figure3(a), the network depth remains fixed, determined solely by the number of layers. However, when a cyclic structure is introduced, as depicted in Figure3(b), the model depth effectively increases with the propagation steps $T$. Specifically, at $T=1$, the output of each layer corresponds to a depth of $1$, as it directly incorporates information from the input. At $T=2$, each layer aggregates two types of information: depth-0 information directly from the input and depth-1 information propagated from neighboring computational neurons, resulting in a maximum depth of $2$. As $T$ increases, the depth of the information available to each layer grows proportionally, enhancing the network's expressiveness. The cyclic structure increases the model's effective depth through iterative propagation, allowing the network to achieve greater expressiveness without additional parameters.”
> >
> >
> > **Q6:** Do the authors really mean Figure 2(b) to be in the current form, or in the edited form elaborated above? For FF-Chain, do the authors mean the current Figure 2(b) or the edited form? What about BP-Chain and BP-Chain*?
> >
> > **A6:** Figure 2(b) illustrates the model structure in Hinton’s paper [1]. FF-Chain is current Figure 2(b), which exactly reflects the model structure in [1]. BP-Chain* is illustrated in Figure 2(a) building the network layer-by-layer and training with global cross entropy loss. BP-Chain uses the structure of Figure 2(b). But the gradient is obtained from cross entropy loss rather than the forward-forward loss.
> >
> >
> > [1] Hinton, G. (2022). The forward-forward algorithm: Some preliminary investigations. arXiv preprint arXiv:2212.13345.
> >
> >
> >
> > Thank you for your constructive feedback, which greatly improves the paper's quality. Hope we have clear all of your concerns. Let us know if more concerns are there :).

---

### Official Review · Reviewer_ZQMN · 2024-11-03

**Soundness:** 2
**Presentation:** 3
**Contribution:** 2
**Rating:** 6
**Confidence:** 4

**Summary:**

The paper introduces Cyclic Neural Networks (Cyclic NNs), a design paradigm that extends neural network computation beyond sequential, layer-by-layer connections. Inspired by the complex, cyclic connectivity observed in biological neural networks, the authors propose allowing neurons in ANNs to form connections in any graph-like structure, including cycles.

**Strengths:**

1.	The paper is well-presented with helpful visualization and a relatively clear description of the method and experiment.
2.	Different shapes of computational graphs are experimented with cyclic NN, which provides insight into how it affects the performance of the model.

**Weaknesses:**

1.	The innovation of the paper is limited. Directed acyclic computation only applies to relatively simple feedforward neural networks. Alternative computational patterns such as recurrent and graph-shaped fall under their respective category (recurrent neural network and graph neural network). The resulting cyclic NN may be succinctly captured with a recurrent type of GNN.
2.	The experimental results and comparisons are limited. The improvement of the approach is only supported in the case of a complete cyclic NN graph. The baseline comparison is limited to a feed-forward network.

**Questions:**

1.	Related to W1, How is GOMLP different from a recurrent graph neural network with the same computational pattern?
2.	Since local learning is used to avoid training on recurrent connections globally, how is the training stability of the model under different graph configurations?
3.	What’s the full asymptotic complexity of the model? In section 3.5., only those relevant to the shape of the graph is discussed.

---

> ### Author Response · Authors · 2024-11-19
> **Rebuttal to Reviewer ZQMN W1, Q1 and W2**
>
> We would like to thank the reviewer for providing constructive feedback on our proposed cyclic neural network. To clarify our strength and make the contribution clearer. We conduct additional experiments to answer the reviewer's questions. Our rebuttals are provided as follows:
>
> **Rebuttal to Weakness 1 (W1) and Question 1 (Q1):**
>
> *Concern:*
>
> The differences between our proposed Cyclic Neural Network (Cyclic NN) and recurrent neural networks (RNNs) or graph neural networks (GNNs) are unclear.
>
> *Response:*
>
> To highlight the distinctive characteristics of Cyclic NN, we have added a new Section A.4 in the appendix, along with Figure 5, which provides a clear illustration comparing Cyclic NN with RNNs and GNNs.
>
> *Comparison with RNNs:*
>
> Recurrent structures like RNNs, LSTMs, and GRUs focus on the recurrence of the same computation block. In contrast, our Cyclic NN emphasizes cyclic communication between different computation blocks, as highlighted in red in Figure 5. The presence of cyclic structures enables us to build neural networks with any graph structure beyond directed acyclic graphs (DAGs). While recurrent structures can be seen as self-loops over a single computation neuron, Cyclic NN allows for more flexible network structures beyond self-loops.
>
> *Comparison with GNNs:*
>
> In GNNs (such as GCNs, recurrent GNNs, and GATs), the graph $\mathcal{G}$ is the input to the network, aiming to learn representations for each node. Typically, DAG-structured computations are used within the model, like the linear layers in GCNs. GNNs serve as encoders for nodes within graphs, with the graph structure acting as the model's input. However, in Cyclic NN, the input is not constrained to graphs; it can be an image, for example, and the Cyclic NN encodes this input into a representation. Here, the graph structure $\mathcal{G}$ refers to the encoder itself within the Cyclic NN.
> Therefore, our Cyclic NN is distinct from both RNNs and GNNs (including recurrent GNNs). The newly added Figure 5 provides a clearer illustration of these differences.
>
>
> **Rebuttal to Weakness 2 (W2):**
>
> *Concern:*
>
>  The improvement of Cyclic NN might not be generalizable across different graph structures.
>
> *Response:*
>
>  The improvements of Cyclic NN are also observed in WSGraph and BAGraph structures. As shown in Table1, the widely used DAG-structured BP-Chain* method achieves an error rate of 1.77 on the MNIST dataset. In comparison, Cyclic NN with WSGraph and BAGraph achieves error rates of 1.70 and 1.64, respectively, both surpassing the current DAG solution. This demonstrates that the improvement of our approach is consistent across different types of Cyclic NN graphs.
> Our core contribution lies in introducing cyclic structures among different computation blocks. To ensure a fair comparison among all training methods, we adopted the same structure as feed-forward networks. To further validate the effectiveness of Cyclic NN, we added an ensemble method that combines multiple linear layers as a baseline. The experimental results, shown in Table 1, indicate that Cyclic NN still performs the best among all methods. This underscores the advantages of incorporating cyclic structures within the model.
>
> | Train        | Graph     | MNIST          | NewsGroup        | IMDB           |
> |--------------|-----------|----------------|------------------|----------------|
> | MLP-Ensemble | -         | 1.91±0.21      | 45.35±0.84       | 17.36±0.23     |
> | BP           | Chain*    | 1.77±0.16      | 42.11±0.92       | **17.16**±0.19 |
> | FF           | Chain     | 1.83±0.20      | 43.88±0.28       | 18.75±0.92     |
> | BP           | Chain     | 1.74±0.11      | 38.85±0.42       | 17.27±0.13     |
> | FF           | Cycle     | 1.80±0.14      | 43.54±0.41       | 18.97±0.49     |
> | FF           | WSGraph   | 1.70±0.17      | 38.28±0.13       | 17.93±0.28     |
> | FF           | BAGraph   | 1.64±0.08      | 38.41±0.14       | 18.20±0.67     |
> | FF           | Complete  | **1.54**±0.05  | **38.266**±0.06  | 17.58±0.20     |

---

> ### Author Response · Authors · 2024-11-19
> **Rebuttal to Reviewer ZQMN Q2 and Q3**
>
> **Rebuttal to Question 2 (Q2):**
>
> *Concern:*
>
>  Clarity on the stability and effectiveness of localized optimization in training.
>
> *Response:*
>
>  To address this, we have added a new Section A.5 in the appendix, which presents the training curves for different graph structures. We plotted the feed-forward (FF) loss, classifier loss, and error rate changes over training epochs. Observations indicate that for all graph structures and datasets, the decrease in losses and error rates is stable and steady. Localized optimization focuses on optimizing parameters at a local level without propagating updates across layers, helping to mitigate gradient vanishing or exploding issues commonly encountered in global optimization.
>
> **Rebuttal to Question 3 (Q3):**
>
> *Concern:*
>
> The need to provide the full asymptotic complexity of the model.
>
> *Response:*
>
> We appreciate this suggestion. In Section 3.5, we have added a paragraph to illustrate the full asymptotic complexity of the proposed GOMLP model:
> "Consider the example of GOMLP and examine the time complexity of each computation neuron. The maximum complexity for each computation neuron is $O((|\mathcal{V}| - 1)d^2) = O(|\mathcal{V}|d^2)$ when it receives information from all other computation neurons. Therefore, the total time complexity of GOMLP is $O(|\mathcal{E}||\mathcal{V}|d^2)$."
>
>
> We hope that these clarifications address your concerns and highlight the contributions of our work more effectively. Let us know if you have any further questions. We would be very happy to improve our paper based on your suggestions :).

---

> ### Comment · Reviewer_ZQMN · 2024-11-25
>
> Thanks for addressing my questions. The rebuttal has clarified most of my concerns. I have adjusted the score.

---

> > ### Author Response · Authors · 2024-11-26
> > **Reminder for Score Adjustment**
> >
> > Thank you for your thoughtful feedback and for taking the time to review my paper. I truly appreciate your acknowledgment that the rebuttal addressed most of your concerns.
> >
> > I noticed that the score associated with your review has not yet been updated. As the review deadline is approaching, I wanted to kindly remind you in case it was overlooked. I understand how busy things can get, and I greatly appreciate your efforts in ensuring the review process runs smoothly.
> >
> > Please let me know if there is any further clarification or additional information I can provide to assist.
> >
> > Thank you again for your valuable time and support.

---

> > > ### Author Response · Authors · 2024-11-30
> > > **Reminder of Score Adjustment of Reviewer ZQMN**
> > >
> > > Dear Reviewer ZQMN:
> > > As we have addressed most of your concerns, we kindly remind you to adjust your score accordingly, as the deadline is approaching.

---

> > > > ### Author Response · Authors · 2024-12-02
> > > > **Reminder of Score Adjustment for Reviewer ZQMN**
> > > >
> > > > As the deadline is approaching today, we would like to kindly remind you to adjust the rating score as indicated in your feedback. Your review and rating are extremely important to us, and we sincerely appreciate the time and effort you have dedicated to evaluating our submission.
> > > >
> > > > Please let us know if there are any issues or further clarifications needed from our side.
> > > >
> > > > Thank you again for your valuable contribution to the review process.

---

### Meta-Review · Area_Chair_Jd7L · 2024-12-19

**Metareview:**

This paper proposes a novel NN framework with an architecture and training paradigm akin Hinton's 2022 forward-forward approach. The empirical results on classical classification benchmarks look very promising. All reviewers have been moved towards favoring acceptance. Looking at the paper I am convinced that it will hurt the paper's impact to publish it in its current form. It is not super clear what is going on and one does not have to dig deep to find inaccuracies. For example, eq (6) is called a cross entropy. Also symbols are used in a pretty non-standard way such as denoting a the output of a function with a relu activation function p and so on.

So the content is worth while accepting but the authors need more time to make this accessible to the scientific community.

**Additional Comments On Reviewer Discussion:**

None.

---

### Decision · Program_Chairs · 2025-01-22

Reject